# Revisiting Austfonna, Svalbard with potential field methods – A new characterization of the bed topography and its physical properties

Marie-Andrée Dumais[1,2], Marco Brönner[1,2]

[1]Department of Geoscience and Petroleum, Norwegian University of Science and Technology, 7031 Trondheim, Norway
[2]Geological Survey of Norway, 7040 Trondheim, Norway

*Correspondence to*: Marie-Andrée Dumais (marie-andree.dumais@ngu.no)

**Abstract.** With hundreds of meters of ice, the bedrock underlying Austfonna, the largest ice cap on Svalbard, is hardly characterized in terms of topography and physical properties. Ground penetrating radar (GPR) measurements supply ice thickness estimation but the data quality is temperature-dependent, comprising uncertainties. To remedy this, we include
airborne gravity measurements. With a significant density contrast between ice and bedrock, sub-glacial bed topography is effectively derived from gravity modeling. While the ice thickness model relies primarily on the gravity data, integrating airborne magnetic data provides an extra insight of the basement distribution. This contributes to refine the range of density expected under the ice and improve the sub-ice model. From this study, a prominent magmatic N-S oriented intrusion and the presence of carbonates are assessed. The results reveal the complexity of the sub-surface lithology characterized with
different basement affinities. With the geophysical parameters of the bedrock determined, a new bed topography is extracted, adjusted for the potential field interpretation, i.e. magnetic and gravity data analysis and modeling. When the results are compared to bed elevation maps previously produced by radio echo-sounding (RES) and GPR data, the discrepancies are pronounced where the RES and GPR data are scarce. Hence, areas with limited coverage are addressed with the potential field interpretation, increasing the accuracy of the overall bed topography. In addition, the methodology improves the
understanding of the geology, assigns physical properties to the basements, and reveals the presence of softer bed, carbonates and magmatic intrusions under Austfonna which influence the basal sliding rates and surges.

## 1 Introduction

During the last few decades, with satellite technology advancement and an increased need to understand climate change, the polar regions have become an important laboratory for studying on-going environmental changes. In this context, icecaps,
icefields and glaciers are of interest as they are highly sensitive to climate variations (Vaughan et al., 2013; Dowdeswell et al., 1997). Glacial sliding and melting rates are often determined from Global Positioning System (GPS) measurements, satellite imagery and satellite altimetry (e.g. Przylibski et al., 2018; Bahr et al., 2015; Grinsted, 2013; Radić et al., 2013; Dunse et al., 2012; Gray et al., 2015; Moholdt et al., 2010b). The ice thickness and the ground topography at the glacier base, key factors to understand the glacial sliding and ice melting mechanisms (Clarke, 2005), have proven challenging to derive.
The glacier deformation mechanisms and sliding depends on the roughness, the rheological properties of the bed, the distribution of the rheological properties of the ice, and the hydrological system at the ice-bed interface (e.g. Gong et al., 2018; Gladstone et al., 2014; Olaizola et al., 2012; Clarke, 2005). Presence of sediments may also contribute to bed deformation resulting in ploughing (basal sliding) (e.g. Eyles et al., 2015; Iverson et al., 2007; Bamber et al., 2006; Boulton and Hindmarsh, 1987; Clarke, 1987). Thus, determining the glacier bed lithology is as critical as determining its topography
to assess glaciers response to climate variations.

Ground penetrating radar (GPR) is the preferred method to retrieve the glacial bed topography, however, scattering from englacial meltwater streams and dielectric absorption often hamper accurate imaging of the bed, especially for temperate ice. For temperatures at pressure melting point, common in temperate glaciers, liquid water is present at the ice-bed interface. The correctness of the resulting topography depends on several glacier parameters, including density, porosity, and water

content fraction, which determined the permittivity and therefore, the radio-wave velocity used to derive the thickness (Lapazaran et al., 2016). These parameters cannot be directly measured and are highly influenced by temporal and spatial variation of the water content fraction distribution through the glacier (Barrett et al., 2007; Navarro et al., 2009; Jania et al., 2005).

Using GPR and radio-echo sounding (RES) measurements from several campaigns, a bed topography has been derived for Austfonna on Svalbard (Fürst et al., 2018; Dunse et al., 2011). In this paper, we test the feasibility to retrieve the glacier thickness of Austfonna with airborne gravity data, as it is sensitive to density contrast between the ice and the bedrock. Adding magnetic interpretation to the study contributes by indicating variations in the bedrock lithology and potential density variations, which must be considered to derive a correct ice thickness and bedrock topography. Combined gravity-magnetic interpretation is a powerful tool to define basement types and identify the presence of various geological structures, such as sedimentary basins under the ice, in the bedrock. Gravity and magnetic methods have been used in the past for basement lithology studies in the Arctic (e.g. Gernigon et al., 2018; Døssing et al., 2016; Nasuti et al., 2015; Gernigon and Brönner, 2012; Olesen et al., 2010; Barrère et al., 2009) and for sea-ice and glacier studies (e.g. An et al., 2017; Gourlet et al., 2015; Tinto et al., 2015; Zhao et al., 2015; Porter et al., 2014; Tinto and Bell, 2011; Studinger et al., 2008; Studinger et al., 2006; Spector, 1966). In this study, we combine them with GPR data to obtain both an accurate glacial bed topography and rheological changes of the basement. Magnetic and gravity modeling were used to assess the feasibility of retrieving topographical and geophysical properties in terms of ice thickness, bed softness, presence of carbonates and till, and bed topography.

## 2 Austfonna and its underlying geology

With a geographic area of 8357 km$^2$, Austfonna, seen on Fig. 1, is the largest ice cap on Svalbard archipelago (Dallmann, 2015). It is located on Nordaustlandet, the second largest island in Svalbard, northeast of Spitsbergen and approximately 80 % of it is covered by ice. Austfonna has one main central dome with an ice thickness of up to 600 m (Dowdeswell et al., 1986) and feeds several drainage basins. Considered polythermal, consisting of a mixture of temperate and cold ice, it is relatively flat at its highest elevation and includes both land-terminating and tidewater glaciers. Studies suggest its basal temperature is near the pressure melting point (Dunse et al., 2011), thus Austfonna experiences basal sliding and subglacial water might be present. Surging, or surge-type, glaciers have also been observed in the area (Schytt, 1969). Other studies link surging to the softness of the bedrock and tectonically active zones (e.g. Jiskoot et al., 2000). The bedrock topography (including cavities and obstacles), geothermal sources and the presence of sediments are also contributing factors to the glacier basal sliding velocities (e.g. Boulton and Hindmarsh, 1987; Clarke, 1987).

During the last few decades, several campaigns aimed to retrieve the underlying bedrock topography of Austfonna using RES (Moholdt et al., 2010a; Dowdeswell et al., 1986) and GPR (Dowdeswell et al., 2008; Dunse et al., 2011). Acquired profiles are shown in Fig. 2. McMillan et al. (2014) and Moholdt et al. (2010a & 2010b) applied satellite altimeter data to estimate surface elevation changes and ice loss. They observed and concluded a significant increase in the dynamic and the outlet of the glaciers Vestfonna (Schäfer et al., 2012) and Austfonna (McMillan et al., 2014). Over 28 % of the area covered by Austfonna rests below sea level (Dowdeswell et al., 1986). Moreover, the lowest elevations of the bedrock are located at the tips of Basin-3, at the southeast, and Leighbreen, at the northeast, (Fig. 1), with bed elevation values of 150 m and 130 m below sea level, respectively (Dunse et al., 2011; Dowdeswell et al., 2008).

The geology underneath the ice is barely understood as very few outcrops are available to identify the main geological structures and basement affinity of Nordaustlandet (Fig. 2). However, based on the studied outcrops, the geology appears to be complex and the exposed rocks are dated to various geological epochs (Dallmann, 2015; Johansson et al., 2002). Basement outcrops at the Wahlenbergfjorden identify different types of basements on each side of the fjord (Dallmann,

2015), which is assumed to represent a major geological N-S division of the island. For the northern shore of the fjord, and north of Nordaustlandet (including the totality of Vestfonna), the regional map of Lauritzen & Ohta (1984) and radiometric dating (Ohta, 1992) indicate a Pre-Caledonian basement with Mesoproterozoic and Neoproterozoic rock exposures, mainly composed of meta sedimentary rocks like marble, quartzite and mica schist. The rocks are significantly folded and faulted

due to the Caledonian deformation influence but, not to the same degree as the rest of Svalbard. Caledonian and Grenvillian Rijpfjorden granites are found on the northern tip of Nordaustlandet, on Prins Oscars Land (Johansson et al., 2005; Johansøn et al., 2002). In the east, the bedrock comprises mainly Silurian diorites and gabbros as seen on Storøya (Johansson et al., 2005). On the south shore of Wahlenbergfjorden, an abundance of Carboniferous to Permian limestones and dolomites are exposed with Early Cretaceous doleritic intrusions. Dallmann (2015) consequently concluded that the

same geological demarcation observed in Wahlenbergfjord continues under Austfonna. The southern basement of Austfonna is believed to be much younger than the one in the north and is composed of unmetamorphosed, post-Caledonian rocks. The youngest rocks in Nordaustlandet are Jurassic–Cretaceous doleritic dikes, which intrude the Tonian basement rocks (composed of dolomite, sandstone, quartzite and limestone) on the island of Lågøya and the Meso to Neoproterozoic basement composed of basalt conglomerate, volcanic breccias and migmatites in the outlet of Brennevinsfjorden, north-west

of Vestfonna (Overrein et al., 2015). South of Nordaustlandet, dolerite sills were emplaced during the Cretaceous in Kong Karls Land. Evidences for the locations of the sills can be found in the seismic reflection and magnetic data in the vicinity of Nordaustlandet (Polteau et al., 2016; Minakov et al., 2012; Grogan et al., 2000).

**3 Magnetic and gravity data**

The magnetic map is a compilation of various datasets flown in 1989 and 1991 (Table 1). The data are sparse with line

spacing of 4 to 8 km at a target ground clearance of 900 m. Having been originally processed by different entities, with different processing algorithms, the dataset is reprocessed to a similar level. A control-line, flown by each survey as an overlap, is used to level the two datasets to each other. This step ensures that the two datasets were leveled to the standard International Geomagnetic Reference Field (IGRF) model (Thébault et al., 2015) and the compilation is smooth at the overlap.

The magnetic map (Fig. 3b) presents strong parallel high anomalies crossing the center of Nordaustlandet oriented N-S. The magnetic intensity is correlated with the type and level of magnetization which in turn is mainly related to the iron content, time of formation or metamorphic processes of the minerals found in the basement. Thus, the magnetization is a strong indicator of the mineralogy of the basement and its lithology. The strong anomaly observed across Austfonna, also intersects the Caledonian Rijpforden granites, which have been identified on the geology map (Johansson et al., 2005). This anomaly is

also parallel to the Billefjorden fault zone and to the Caledonian Frontal Thrust (Gernigon and Brönner, 2012; Barrère et al., 2009). The Caledonian is also associated with magmatic episodes. Northeast of Nordaustlandet, the sharp and low-frequency magnetic anomalies created by the known emplaced Cretaceous sills have a distinct and prominent signature.

The gravity data were acquired during a campaign in 1998-1999 (Forsberg and Olesen, 2010; Forsberg et al., 2002). The lines were flown SW-NE with a spacing of 18 km and at a ground clearance of 1 km (Table 1). The free-air anomaly map is

presented in Fig. 3a. The gravity data produced 4000 m cell-size grids with a standard deviation of ~2 mGal over 6000 m half-wavelength resolution. Gravity lows are seen on the south and southwest of Nordaustlandet with a higher signal on the ice cap reflecting the ice coverage and its thickness. Gravity is sensitive to the density contrast between the various geological bodies, and ice in this case. Low gravity measurements reflect low densities, which are often linked to sediment accumulation or sedimentary basins.

The grid resolution provides an estimate of the level of smoothness of the data and of the limitations for the modeling and data filtering. Given the magnetic grid resolution, features shallower than 2 km cannot be accurately resolved. Depths

interpretation and body geometry are limited by the grid resolution. A single anomaly normally leads to several geometry and depth possibilities. In this paper, the most favorable possibility is chosen for its consistency with the GPR and RES investigations and the model simplicity. Therefore, depth estimates from the models in the present paper represent the deepest depth possibility and limited by 2 km resolution. The magnetic data also present several asymmetric anomalies

which can be interpreted by dipping bodies. However, given the coarseness of the data, a simple model without dipping is preferred.

## 4 Bed topography revisited

Dunse et al. (2011) have presented a bedrock topography compilation with 1 km spatial grid resolution from data acquired by RES and GPR, but the geospatial distribution of the measurements (Fig. 2) suggest lower resolution in areas with poor

coverage. With these data, combined with the ice surface topography published by the Norwegian Polar Institute (NPI) in 1998 (Norwegian Polar Institute, 1998), an ice thickness is derived for Austfonna. This step allows an estimate of the volume and mass of the ice cap to derive the gravitational effect of the glacier. The density contrast and the topography of the bedrock-ice interface contributes to the sharpest and most prominent gravity effects. A valid approach to resolve the bedrock topography is to assume a simple basement geometry, with a homogeneous density. Analogous to sedimentary basin

interpretation (Bott, 1960) and treating the glacier as an infinite slab, the free-air anomaly ($FA_c$) along a profile is reconstructed:

$$FA_c = 2\pi G \rho_{ice} h_{ice} + 2\pi G(\rho_{bed} - \rho_{ice})H_{ice} + 2\pi G \rho_{bed} h_{bed} \ , \tag{1}$$

where G is the gravitational constant ($6.67 \times 10^{-11}$ N m$^2$ kg$^{-2}$), $\rho$ the density, $h_{bed}$ the topography of the bed above sea-level and $h_{ice}$ and $H_{ice}$ the thickness of ice above sea-level and below sea-level, respectively. ($h_{ice}$+$H_{ice}$) represents the full extent of

the ice thickness. The free-air anomaly is referenced to the geoid. In the reconstruction of the free-air anomaly, the ice above sea-level is regarded as an excess of mass whereas, the ice below sea-level is considered a mass deficiency. The influence of the ice ($\rho_{ice}$ = 910 kg m$^{-3}$) depends on the surrounding medium, which include air ($\rho_{air} \approx 1$ kg m$^{-3}$, negligible) and the bed ($\rho_{bed}$ = 2670 kg m$^{-3}$), in this case. This reduction technique is valid under the condition that the thickness of the ice is smaller than the horizontal dimensions of the ice cap by several magnitudes. As GPR and RES were acquired onshore solely, gravity

acquisition only onshore was considered in the model for comparison.

Assuming the difference between the free-air anomaly observed ($FA_o$) and the free-air anomaly calculated ($FA_c$) is caused by erroneous bed topography measurements, the correction of the bed topography is:

$$\partial h_{bed} = \begin{cases} \frac{(FA_o - FA_c)}{2\pi G(\rho_{bed} - \rho_{ice})}, \text{ if the bed topography is below sea-level} \\ \frac{(FA_o - FA_c)}{2\pi G(\rho_{bed})}, \text{ if the bed topography is above sea-level} \end{cases}, \tag{2}$$

On average, this correction is 2 m for the analysis along the gravity profiles above Austfonna. With a standard deviation of

63 m, the difference in thickness varies between -190 m to 290 m. The difference in thickness is applied to the initial bedrock topography derived from GPR/RES. Given the wide line spacing of the gravity profiles, both datasets are gridded with the same resolution (4000 m) for the analysis (Fig. 4). The highest summits of the bed topography remain at the same level. Small residual discrepancies are largely due to the approximation of an infinite slab and the accuracy of the various datasets. It should be noted that both GPR/RES depth measurements and gravity ice thickness were calculated with the same

ice surface topography dataset which acts as a control variable. It reduces the influence of the resolution and accuracy of the ice surface topography when comparing the two bed topography models. However, important discrepancies exist, for e.g., under Vegafonna, the southwest corner of Austfonna, and under Leighbreen and Worsleybreen, northeast of Austfonna. These areas are discussed in detail in later sections where magnetic data are included in the interpretation. Less prominent misfits occur at the outer edge of the marine-terminating glaciers Basin-3 and Bråsvellbreen where the ice surface

topography and glacier geometry might undergo rapid and drastic variations and relatively faster ice surface velocities were

observed in comparison to the thick, flat interior ice cap (Gladstone et al., 2014; Moholdt et al., 2010a). As the ice surface topography and gravity data have been acquired around the same time but independently, the resolution and accuracy of the ice surface topography would increase the misfit where the glacier geometry is most susceptible to drastic variations. Notably, the gravity profiles cross the glacier perpendicular to its flow, parallel to the shore edge with an uneven mass

distribution, i.e. more mass is found on the northern side of the profile. This terrain effect is commonly corrected for in gravity processing, for extreme topography relief (Lafehr, 1991), but requires accurate terrain topography such as laser scanning data acquisition or a high-resolution digital elevation model.

## 5 2D-forward models

2D-forward modeling interpretation determines the interface between contrasting bodies, of different magnetizations and

densities. It provides depth and geometrical insights into lithological variations in the bedrock. The forward modeling (Fig. 5) is carried out along the actual airborne gravity lines to ensure the highest resolution of the gravity data. Two lines are modeled and referred as profiles A and B (Fig. 2). The modeled profiles were chosen for their location and coverage. They contain several aspects of the geology under Austfonna (basements, intrusions) and they are located near or above RES and GPR measurements. Models are initially constrained by the bedrock topography derived by Dunse et al. (2011) and

independent of the free-air bed topography corrections. The measured data points from the GPR/RES are highlighted (purple circle, Fig. 5a).

Initial petrophysical parameters are assigned based on the comprehensive petrophysical database from the mainland of Norway (Olesen et al., 2010) provided by the Geological Survey of Norway (NGU) and the described bedrock types (Dallmann, 2015). Using gravity and magnetic signatures, the basement is forward modeled accordingly, using the software

package GM-SYS (Geosoft, 2006). The mantle-crust boundary, Mohorovičić discontinuity (Moho), was set at around 33 km depth following the interpretation from Ritzmann et al. (2007). On the northeast of Austfonna, the basement seems to have a very low magnetization (less than 0.001 SI), but a density higher than the surrounding (2700 kg m$^{-3}$) is required to fit the observed field.

On Profile A (Fig. 5a), reducing the density on the southwest of Austfonna (where Vegafonna is located) was attempted but

could not be fit to the observed free-air anomaly. Introducing layers of till, with a density of1600 kg m$^{-3}$, did not significantly reduce the signal to account for the observed gravity data unless the till had a thickness of several hundreds of meters. Thus, the GPR/RES bed topography is adjusted in this area to be consistent with the gravity measurements. This discrepancy is more important under a region with scarce GPR/RES measurements (indicated with purple circles). A similar interpretation was made on profile B, where misfits between the two methods occur and only a few measurements exist. The GPR/RES

data were not acquired in a grid pattern and therefore the GPR/RES bed topography proposed in these discrepancy areas is the result of a gridding interpolation between profiles and data points. The bed topography calculated from the free-air analysis and interpreted from magnetic and gravity modeling agree in general and suggest corrections to the GPR/RES in the same direction. However, misfits exist since the free-air analysis presented in the previous section considers a homogenous basement while the interpretation indicates variable densities. The difference between the bed topography from the

GPR/RES and the 2D-forward model varies from -170 m to 80 m with a standard deviation of 40 m. A smaller level of correction is required with the 2D-forward model than predicted from the gravity correction. The 2D-forward model accounts for a certain degree of confidence on the GPR/RES data and for the bedrock density variation.

The centers of both profiles are characterized by a high magnetic anomaly requiring high susceptibility. This anomaly is a prominent and continuous N-S oriented anomaly, which might at least be partly linked to exposed granites on Prins Oscars

Land at the northern tip of Nordaustlandet. A relatively high density of 2725-2750 kg m$^{-3}$ is assigned to this granitic intrusion. However, granites with comparable densities and susceptibilities are found on the mainland of Norway in Vest-

Adger, Rogaland and Telemark (NGU petrophysics database available at http://geo.ngu.no/GeosciencePortal/, 2016). Werner deconvolution (Phillips, 1997; Ku and Sharp, 1983; Werner, 1955), an automated depth-to-source estimation method, was applied to help quantify the depth and morphology of magnetic bodies under Austfonna (Fig. 5a and 5b). Using these empirical basement indicators that are sensitive to susceptibility variation and approximating the geological source to a simplified geometry such as contacts and dikes (Goussev and Peirce, 2010), the depth and edges of intrusions were estimated. Euler deconvolution (Thompson, 1982; Reid et al., 1990), with a structural index of 1 (dike, sill), was also used to compare the results. This method uses horizontal and vertical derivatives along with a pre-determined structural index to estimate the source location. In our case, Euler deconvolution analyses provides similar depth values to Werner deconvolution. Both Werner and Euler deconvolution analyses determined a dike at a depth of about 8 km with a width of almost 20 km. While Euler deconvolution results into a dike seated at 8 km, Werner deconvolution resolves the top of this intrusion to be tilted with a depth from 8 km on the southwest to 6 km on the northeast. A second dike was determined at 2 km depth (or 1.5 km with Euler deconvolution) with a much narrower width of 2 km and was only seen on Profile A, indicating also a narrower dike in length. The model suggests shallow magnetized bodies offshore Nordaustlandet with a depth of less than 2 km. These indications are used in the model to constrain the depth of the intrusions. Given the accuracy of the data, a certain degree of freedom is allocated to those indicators to fit the observed data with the geology expected.

The gridded tilt derivative of an anomaly, at a location x, y (Miller and Singh, 1994), characterizes the angle of the ratio between the amplitudes of the vertical derivative and the horizontal derivative. Thus, the zero-contour indicates the border of a geological body, where a density or susceptibility contrast with the surrounding occurs. This indication from the magnetic tilt derivative (Fig. 6) was used to constrain the lateral extent of the intrusions. Blakely et al. (2016) have also developed a method to retrieve the edge of a body and its depth (Fig. 5) by using the reciprocal of the horizontal gradient at the zero contour of the tilt derivative grid (Fairhead et al., 2008; Salem et al., 2007). At high magnetic latitude for a vertical dike geometry, the depth is estimated equal to the half-width of the magnetic anomaly (Hinze et al., 2013). The lateral edge of the body is adjusted accordingly to the depth found with Blakely's method (2016). This reduces the size of the magnetic body (Fig. 5) to the minimum size required for this depth. Thus, a first magnetic body with a susceptibility 0.004 SI in a 0.003 SI surrounding, a density of 2670 kg m$^{-3}$, a width of 3 km and a depth of 2 km is modeled. The top of the second intrusion is deeper (10 km) and wider (15 km) with higher magnetic and density properties (0.016 SI and 2750 kg m$^{-3}$). For both profiles, the difference between the bed topography from the magnetic/gravity interpretation and the gravity estimation is caused by the large density intrusion located in the basement.

On profile B, anomalies of smaller sizes are found on the east coastline of Austfonna. The nature of the magnetic signal and the results from Euler and Werner deconvolutions suggest shallow magmatic bodies such as sills. For simplification, they were modelled with a common magnetization value for sills of 0.15 SI susceptibility (Hunt et al., 1995). Another major difference between the two profiles modeled is the higher density body (2840 kg m$^{-3}$) located west of the intrusions on profile B. The NPI geological map identifies a carbonate outcrop in this area of Austfonna. This carbonate body has a strong influence on the gravity signal, which is critical in the estimation of the bed topography (turquoise topography, Fig. 5b). Locally, the bed has a much higher density and should be considered for bed-topography corrections. The magnetic and gravity modelling provides an indication of this carbonate depth, orientation and thickness provided. Given the coarseness of the data and their limitations, the carbonate body is expected to be shallower and thinner.

**6 Bed lithology revisited**

The results from the 2D modeling of profiles A and B are summarized in Fig. 7. According to the models, a prominent deep-seated high-magnetic intrusion occurs underneath Austfonna crossing N-S and the bedrock is divided into two types of basement with different geophysical properties.

Given the densities and susceptibilities used and the presence of granites on the northern part of the island, the intrusion is likely to be granitic. It is probably of a different composition than the exposed rocks since the modeled granitic densities indicate relatively high values but within the expected values for granites (2500-2810 kg m$^{-3}$; Telford et al., 1990) Moreover, given the north-south trending faults system across Svalbard, a similar process could explain the strong magnetic anomalies trending north-south and crossing Nordaustlandet. Major faults on Svalbard, trending N-S to NNW-SSE, have been reactivated and juxtaposed by strike-slip motion, over several geological periods, before, during and after the Caledonian Orogeny (Dallmann, 2015). Granites have been emplaced during the late stages of the Caledonian (Late Silurian to Early Devonian) (Dallmann, 2015). One could argue the presence of N-S striking sills in the near offshore could correlate to the magnetic signature seen under Austfonna. However, the frequency content of the magnetic signal (derived from high-frequencies filters or vertical derivatives), the size of the structures revealed from tilt derivative signal and the depth estimates from Werner deconvolution suggest a rather wide (15 km) deep-seated (10 km) dike intrusion or dike complex onshore and shallow bodies offshore and at the coast line of Austfonna. In addition to the granite affinities suggested by the susceptibility and density interpreted in the 2D-forward models, the high magnetic anomalies correspond to the geological mapping of the observed granites (Fig. 7). Therefore, granite intrusions are proposed in Nordaustlandet bedrock, such as the Caledonian Rijpfjorden granites seen on the Central-northern tip of Nordaustlandet (Johansson et al., 2005). These intrusions, trending N-S to NNW-SSE like the major faults found on Prins Oscars Land, suggest the faults are present and continuing under Austfonna.

On Profile B, sill intrusions are modeled on the east coastline, where shallow sills have been previously interpreted and related to a tholeiitic phase (130-100 Ma) linked to the spreading of the Amerasia basin in the Arctic Ocean and the uplift initiated by the mantle plume on the Yemark plateau, northwest of Svalbard (Polteau et al., 2016; Minakov et al., 2012; Grogan et al., 2000).

Lower densities are also found under southwest Austfonna compared to the northeast region. This is consistent with the terrain observations under Etonbreen and Bråsvellbreen basins (Dunse et al., 2015; Dunse et al., 2011), suggesting a more erodible bedrock in the southwest area. It correlates with the Late Paleozoic platform composed of limestones, dolomites, carbonate rocks and sedimentary rocks to the southwest of Austfonna compared to the meta sedimentary rocks (marble, quartzite and mica schist) from the pre-Caledonian basement found to the northeast. Furthermore, the 2D-model suggests a smoother bed topography than the one suggested by GPR/RES measurements which is consistent with a more erodible basement. While two types of bedrock are already expected from outcrop samples, the analysis of the two profiles suggests an oblique division (NE-SW) between the two basement types rather than a N-S division. The younger basement is more constrained to the northeast of Austfonna than previously thought. An oblique division of the basement is consistent with the major fault system found on Svalbard, and geological provinces division, often separated by faults, both trending N-S to NNW-SSE (Harland et al., 1974; Flood et al., 1969).

**7 Methodology assessment**

Additional magnetic and gravity data improve the bed accuracy and the spatial resolution by filling GPR/RES gaps. Austfonna bed topography was assessed and recalculated using free-air anomaly measurements. From the 2D-model interpretation, the bed topography was enhanced and refined, and its physical properties were extracted. Given the low and scarce sampling of the GPR/RES data under Vegafonna, the discrepancies might be due to gridding interpolation (Fig. 3) as previously discussed. Similarly, on profile B, the poorer fit of the bed topography derived from GPR/RES with the magnetic and gravity model is caused by the scarcer availability of GPR/RES data. Another source of error is the accumulation of water in the erodible basement causing an increase of uncertainty and underestimation of the ice thickness. The magnetic and gravity data provide a consistent and regular coverage over the full area and are less sensitive to gridding interpolation.

Gravity data processing requires high precision GPS positioning which was estimated as 0.5 m vertical accuracy (Forsberg et al., 2002). In comparison, the GPR/RES measurements distribution shows irregularities mainly due to the poor navigational guidance available at the time of acquisition (Dowdeswell et al., 1986; Glacier Thickness Database, 2019). The navigational errors caused flight line distortions and wider line spacings in certain areas. Positional errors were estimated as ± 250 m (Dowdeswell et al., 1986). Therefore, the GPR/RES bed topography is more prone to gridding interpolation artefacts. GPR/RES are susceptible to thickness errors in presence of steep bed slope, where the signal is reflected from a lateral wall instead the bottom topography. While often corrected with 2-D migration processing technique correcting for the direction of profiling, transversal slopes are not corrected unless 3-D migration is used (Lapazaran et al., 2016; Moran et al., 2000). The water content in the glacier and the bedrock increases internal scattering and the dielectric absorption. It also affects the radio-wave velocity which contributes to the error in the time-to-thickness conversion (Brown et al., 2017; Lapazaran et al., 2016; Blindow et al., 2012; Matsuoka, 2011). Temporal and spatial variations of radio-wave velocity account for uncertainties in ice thickness reconstruction (Jania et al., 2005; Navarro et al., 2014). The magnetic and gravity interpretation compensates indirectly for these errors as it is less sensitive to water content in the bedrock and offers an additional control on the properties of the bedrock. The magnetic data show the bedrock heterogeneity, associated with susceptibility variations within the glacier bed, indicating different bedrock types and lithologies. These lithological changes suggest the presence of geological boundaries and provide constraints to assign density changes. Thus, the magnetic data improve the final bed topography accuracy as it provides constraints for the density distribution for the bed underlying the glacier. The effect of geology on gravity inversion for glacial bed topography was also noticed in other studies (An et al., 2019, 2017; Hodgson et al., 2019).

Using Austfonna bed topography and lithology derived from the 2D-forward model, the theoretical gravity response was modeled for ice loss by removing iteratively uniform and homogeneous layers of ice (Fig. 8). The model predicts that an ice thickness variation of 10 m causes an average variation in gravity of ~0.5 mGal which is resolved by state-of-the-art gravity measurements. Thus, the gravity anomaly is mainly driven by the bedrock topography and its physical properties, providing hard evidence of the interface between the ice and the rock. The cell size of the GPR/RES gridded bed topography is 1000 m, with extensive interpolation between the measurements. Flown in 1998-1999, the gravity data produced 4000 m cell size grids with a standard deviation of ~2 mGal over 6000 m half-wavelength resolution. Bed topography corrections with gravity data are more effective than GPR/RES gridding interpolation algorithms. The spatial resolution of airborne gravity measurements depends on the gravimeter together with the platform stability, line spacing, acquisition speed and distance to the source. State-of-the-art fixed-wing airborne gravimeter, flown with the appropriate flight parameters can produce 200 m cell size grids with a precision of ~0.5 mGal over 3000-4000 m half-wavelength resolution (e.g. An et al., 2019, 2017; Studinger et al., 2008). Therefore, using gravity modeling increases the confidence and the accuracy of the bedrock topography under a glaciated area. Improvement of the spatial resolution of the final bed topography could also be achieved with the appropriate survey parameters and a denser line spacing for the gravity data.

Till, commonly found at the base of the glacier, can account for the misfit between the observed and modeled gravity but could not be resolved given the resolution of the dataset. For a variation of 1 mGal, 50 m of till (1600 kg m$^{-3}$) needs to be emplaced in the model. Lower flight elevation and denser line spacing acquisition is required to model the till. For accurate interpretation till modeling, additional independent measurements are required, such as magnetic data which is sensitive to the susceptibility contrast with the surrounding bedrock.

Due to their chemical composition, calcium carbonate rocks erode sub-glacially and migrate in the glacier system through various transportation paths (Bukowska-Jania, 2007). Calcite dissolution and precipitation chemical processes have a role on the calcite saturation of the water film that lubricates the bed-glacier interface and modify the bed morphology and roughness through melting and regelation processes (Ng and Hallet, 2002). The model from profile B suggests carbonate rocks underlying the glacier and maps the lateral extent of the body by an additional 7-8 km under the ice. While the

thickness of the carbonate is small compared to the resolution of the data, the gravity measurements suggest an important excess of mass at that location but no susceptibility variation from the surrounding. Therefore, we must expect a prominent volume of carbonate with an assumed density of 2840 kg m$^{-3}$.

Deep intrusions, possibly granites, and shallow sills were located and delineated from the 2D-forward model. Characterization of these intrusions provides information about the potential variation of the bed lithology in terms of thermal conductivity. Geothermal heat flux, resulting from the decay of radioactive isotopes present in the glacier bed, may raise the temperature of the basal ice and affect the ice sliding (Paterson and Clarke, 1978). Granites are prone to higher geothermal heat flux due to their mineral composition. On Austfonna, only one borehole has been drilled to reach the bedrock and provide heat flux information in the summit area indicating a geothermal heat flux of ~40 mW m$^{-2}$ (Ignatieva and Macheret, 1991; Zagorodnov et al., 1989). There are no other direct observations available to estimate this heat flux. However, our study suggests that this measurement may consequently not be representative for the entire bed underlying Austfonna.

Located outside the 2D-modeled profiles, a high-intensity anomaly is apparent under Basin-3, subject to high negative ice surface elevation change rate (Gladstone et al., 2014; Moholdt et al., 2010a). Due to the large variation of the ice surface topography in the recent years (and decades), retrieving a valid ice topography for the gravity model has proven difficult. However, the results from profile A and B with the Euler deconvolution, Werner deconvolution and Blakely depth methods indicate a basement, resembling to the softer southwest basement, largely intruded with shallow (less than 2 km) and deeper (8 km) magnetized bodies, sills and granitic intrusions respectively. Such physical properties are possible drivers for the high basal sliding rate and surge mechanisms and can be linked to the high ice surface elevation changes seen on Basin-3. Further studies of the granitic intrusion and thermal modeling are of great interest to link the geothermal flux under Basin-3 to ice changes currently observed.

The interpretation of the two profiles provides an insight into the basement and intrusion geology, and a refined glacial bed topography, specifically where GPR/RES data are scarce and less reliable. These findings enhance the understanding of the regional geology of the area and demonstrate the potential to reconstruct the full bed lithology with the aid of high-resolution gravity and magnetic data. Granitic intrusions are known to be potential geothermal sources and can locally affect the heat flux profile of Austfonna. These intrusions can be linked to basal sliding of Austfonna and lead to a better understanding of the sliding mechanisms in the area.

## 8. Conclusion

Airborne magnetic and gravity data were used to study the Austfonna icefield basement on Svalbard. Considering a homogenous basement, the GPR/RES bed topography was corrected for gravity measurements. We demonstrated the importance of the geology for a gravity inversion to calculate the bed topography and presented a method that integrates magnetic, gravity and GPR/RES data. Several interpretation techniques (Euler deconvolution, Werner Deconvolution, 2D modeling) were used to create a model of the bedrock with assigned physical properties in terms of size, depth, susceptibility and density. These results suggest the bed topography derived from GPR/RES measurements can be corrected with gravity analysis while knowledge of the basement lithology and/or magnetic interpretation further increases its reliability. Thus, the bed topography model derived from magnetic and gravity contributes to a more accurate estimation of ice volume. One of the main challenges is that the data were acquired in different campaigns, on different years and with different acquisition patterns. On the other hand, it expands the coverage of the model. Given the difficulty to access the underlying lithology of Austfonna, increasing the magnetic and gravity coverage is an effective method to assess the physical properties of the basement.

Moreover, the geophysical interpretation provides insight into the geological and structural affinity of the basement under Austfonna. While the presence of two basement types on Nordaustlandet is well accepted, the new interpretation allows the boundary between the basements to be mapped. The physical properties of the basements provide indications of the basement types for softness and erodibility and provide information about the type of intrusions likely found under the icefield. Sills, granitic intrusions and carbonate rocks have been interpreted in the model and their evolution was set in a geotectonic time frame. Each of these geological bodies have a different impact on the thermal basal regime and the erodibility of the basement consequently leading to heterogenous basal ice sliding rates.

The temperature of the ice at the base, which controls the basal thermal regime, is usually determined by ice thickness, ice advection, ice surface temperature, geothermal heat and frictional heat (related to softness and topography). Irregular basal topography leads to complex localized patterns of the thermal regime. The lithology identified with potentially higher radiogenic heat production can be correlated with areas of faster ice surface velocities or ice thickness variations. Here, with additional petrophysical properties from collected rock samples, thermal modeling is necessary and will help to better understand the different geothermal domains and its consequences on Austfonna basal thermal regimes. In this paper, the resolution of the datasets limits the resolution of the geometry of the geological features modeled. Higher resolution data from state-of-the-art instrumentation, i.e. gravimeters, GPS, GPR/RES and magnetometers, would further refine the physical properties of the basement and allow a full reconstruction of the bed lithology and topography.

## 9 Data availability

Bed elevation data are available from Dunse et al. (2011). Magnetic and revised bed topography data are available on the Geological Survey of Norway Geoscience Portal (http://geo.ngu.no/GeosciencePortal/search, contact: Marie-Andrée Dumais). Gravity data are available upon request from the Norwegian Mapping Authority (contact: Ove Omang) or on the Geological Survey of Norway Geoscience Portal.

## 10 Author contributions

MAD reprocessed the airborne magnetic dataset and produced the bed topography from the gravity data and the 2D-forward model. MB assisted in the data interpretation and commented on the paper.

## 11 Competing interests

The authors declare that they have no conflict of interests.

## 12 Acknowledgements

We would like to thank Thorben Dunse for providing the GPR/RES bed and related surface elevation maps, and for his helpful discussions on the data. We thank Rene Forsberg and Ove Omang for providing the airborne gravity data. Chantel Nixon is also acknowledged for English proof reading. We thank Daniel Farinotti and three anonymous reviewers for their valuable comments that improved the manuscript.

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

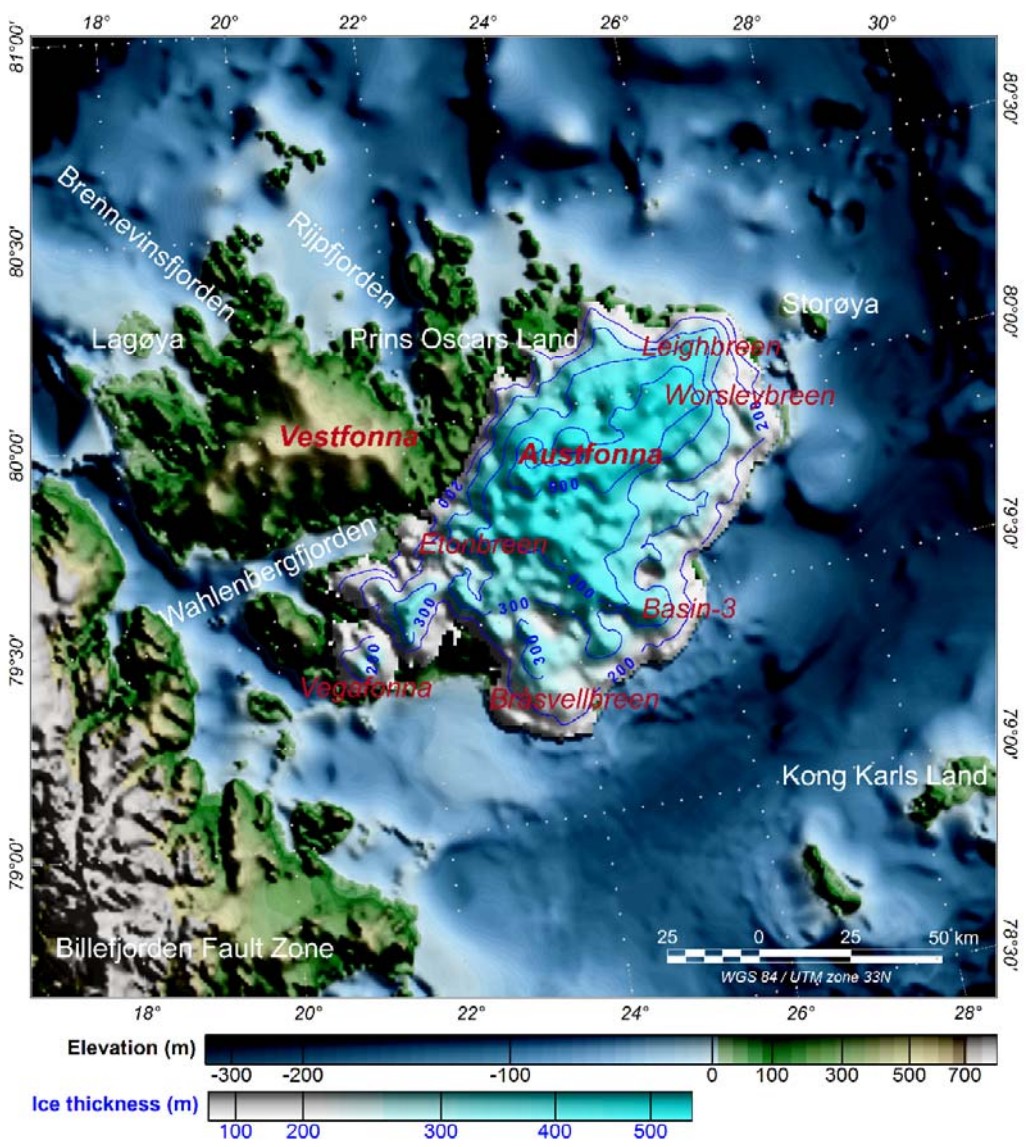

**Figure 1: Surface topography map of Nordaustlandet, east of Spitsbergen, Svalbard, and the Austfonna Ice Cap from Dunse et al. (2011). Approximately 80 % of Nordaustlandet is covered by ice, and an ice thickness up to c.600 m. Polythermal and relatively flat at its highest elevation, Austfonna hosts both land-terminating and tidewater glaciers of which several have been observed to surge.**

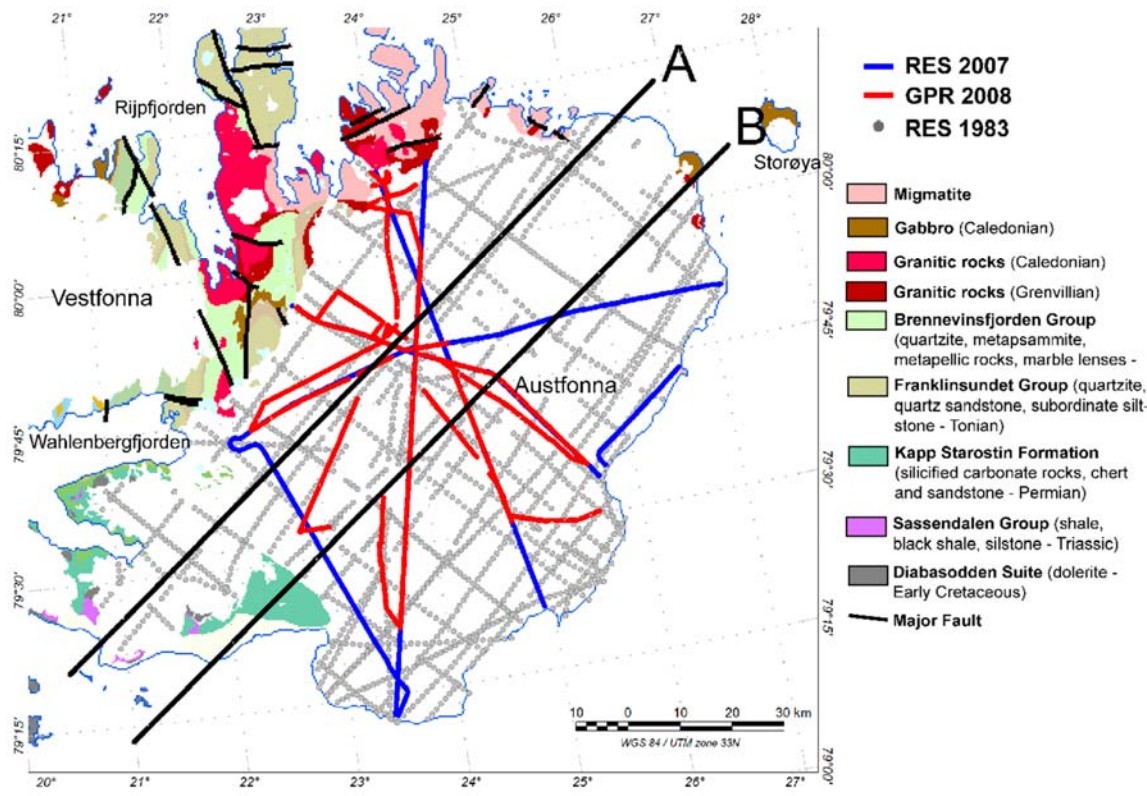

**Figure 2: Geological map of Austfonna with GPR, RES and gravity/magnetic profiles A and B (modified from Dallmann, 2015 and Dunse et al. 2011). The interpreted profiles, labelled A and B, have been chosen to cover a large area of Austfonna and to capture important geological trends.**

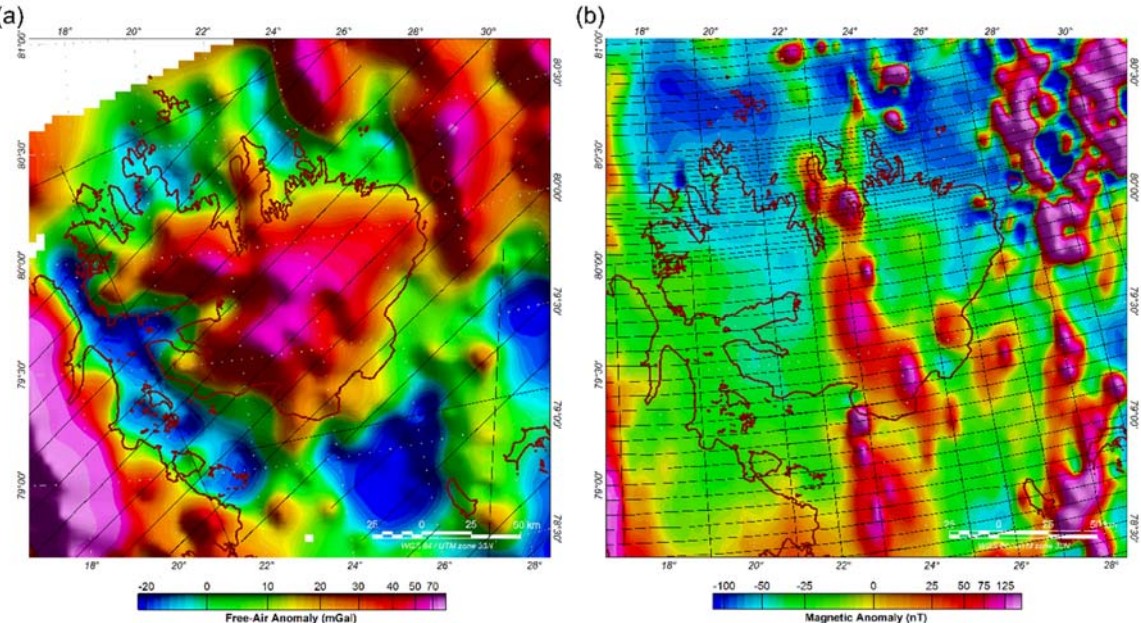

**Figure 3. a) Free-air gravity and b) magnetic anomaly of Nordaustlandet with the acquisition flight lines denoted by the thin black lines. The gravity data are sensitive to an excess or loss of mass. Low free-air gravity data are often linked to sedimentary basins. The magnetic data show important N-S trending anomalies crossing Nordaustlandet and intersecting with the Caledonian Rijpfjorden granites.**

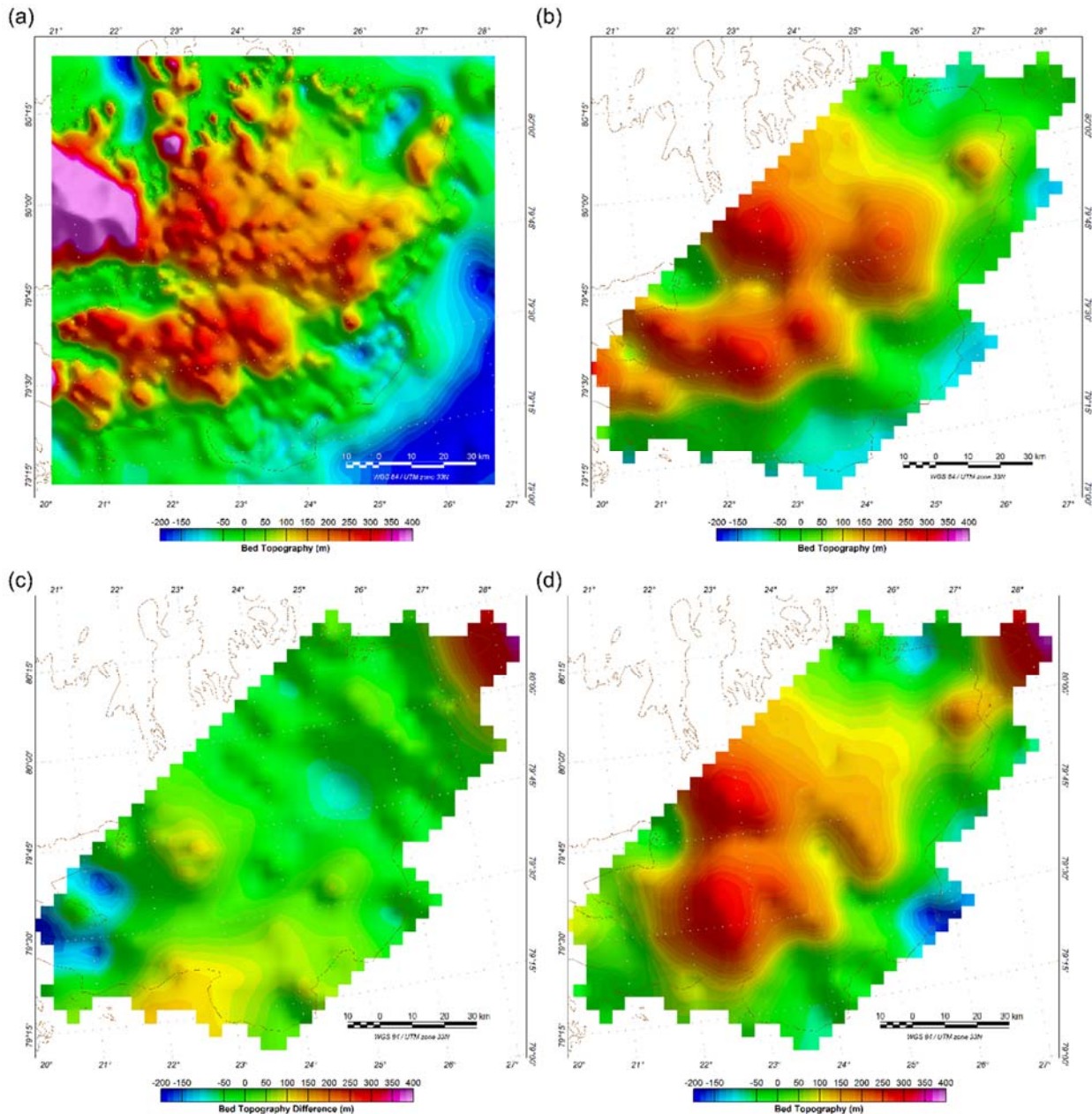

**Figure 4: Bed topography derived from RES and GPR (a), bed topography derived from RES and GPR gridded along gravity acquisition flight lines (b), corrections applied to the bed topography (c) and bed topography corrected for gravity measurements gridded along gravity profiles (d). Major discrepancies, with deviations greater than 150 m, occur under Vegafonna (southwest) and Leighbreen/Worsleybreen (northeast).**

# (a) Profile A

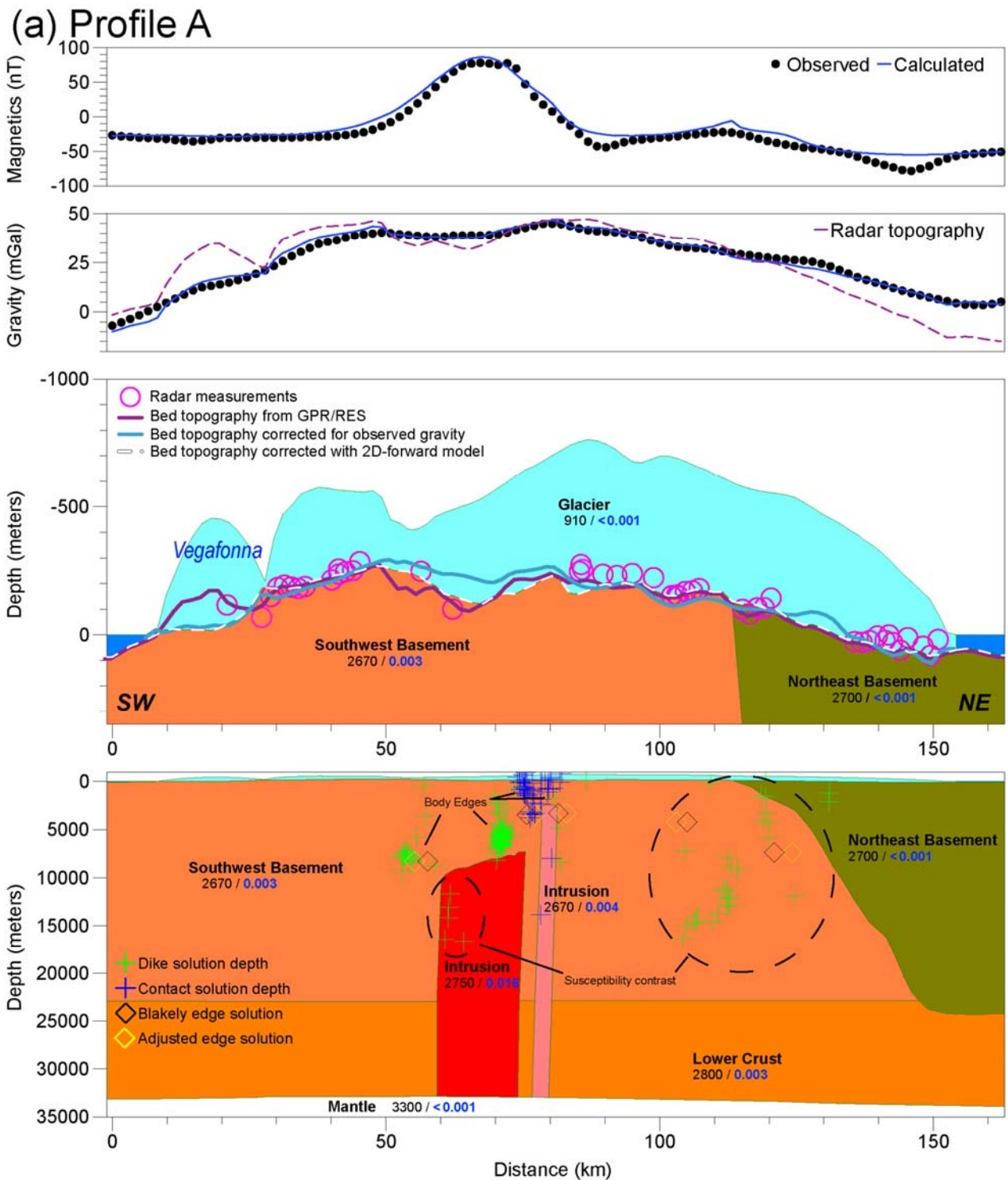

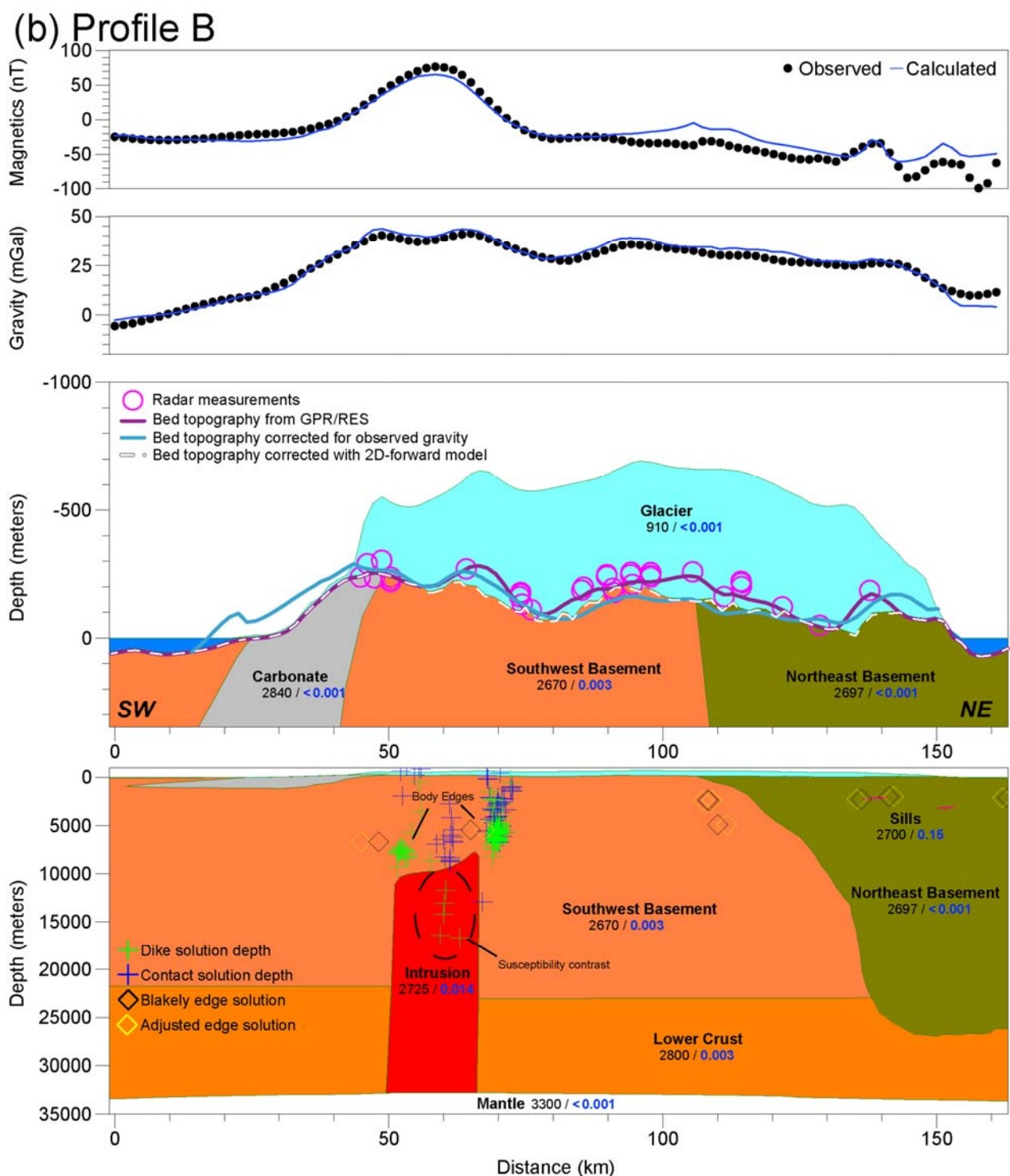

**Figure 5: As defined in Fig. 2, profile A (a) and profile B (b), near-surface view of the basement (top) and depth to mantle (bottom) with Werner deconvolution indicators of the intrusions and the basements interface. A gravity response (purple) is calculated for a homogeneous bedrock using GPR/RES bed topography. The misfit with the observed gravity suggests the bedrock is heterogeneous and the bed topography from the radar needs refining. The gravity corrected bed topography improves but fails to recognize the heterogeneity of the bed. The 2D-forward model improves the accuracy of the bed topography by using a density more representative of the lithology. Each polygon representing a geological body is characterized with a density (kg m$^{-3}$; black) and a susceptibility (SI units; blue). Dike solution depths (green crosses), contact solution depths (blue crosses), Blakely edge solution depths (yellow diamonds) and adjusted edge solution depths (black diamonds) are identified.**

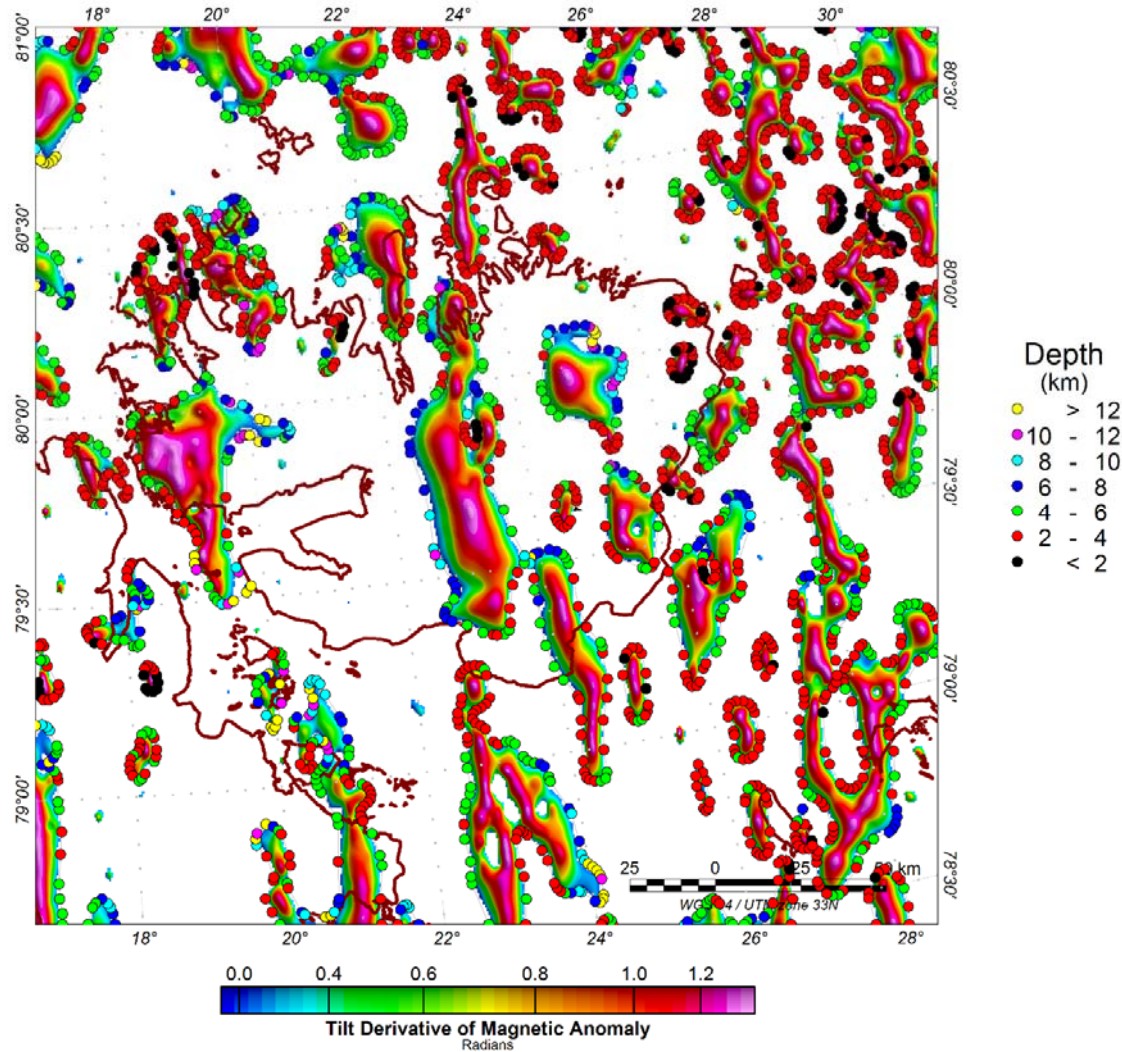

**Figure 6: Tilt derivative of magnetic providing body lineaments, superimposed by Blakely depth estimation, used to determine the location and depth of geological bodies and to constrain the model. Negative data are nulled. Sill bodies located on the N-E of Austfonna, onshore and offshore, are generally shallower than the large and deep N-S trending granitic intrusions crossing Austfonna.**

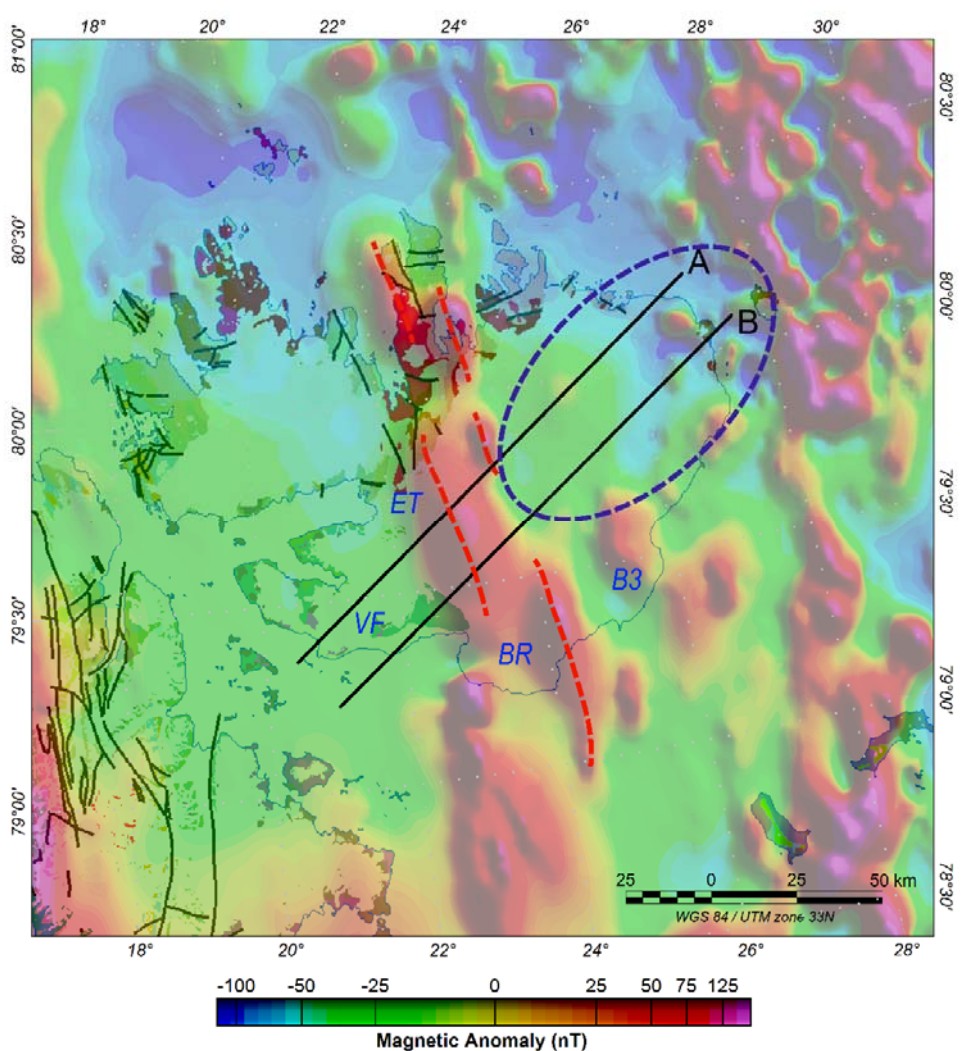

**Figure 7: Profiles A and B modeled, as defined in Fig. 2, against the magnetic anomaly and NPI geological map. Red dashed lines: deep intrusion trends across Austfonna; Blue circle: change of basement seen on the lines modeled (B3: Basin-3; ET: Etonbreen; BR: Bråsvellbreen; VF: Vegafonna)**

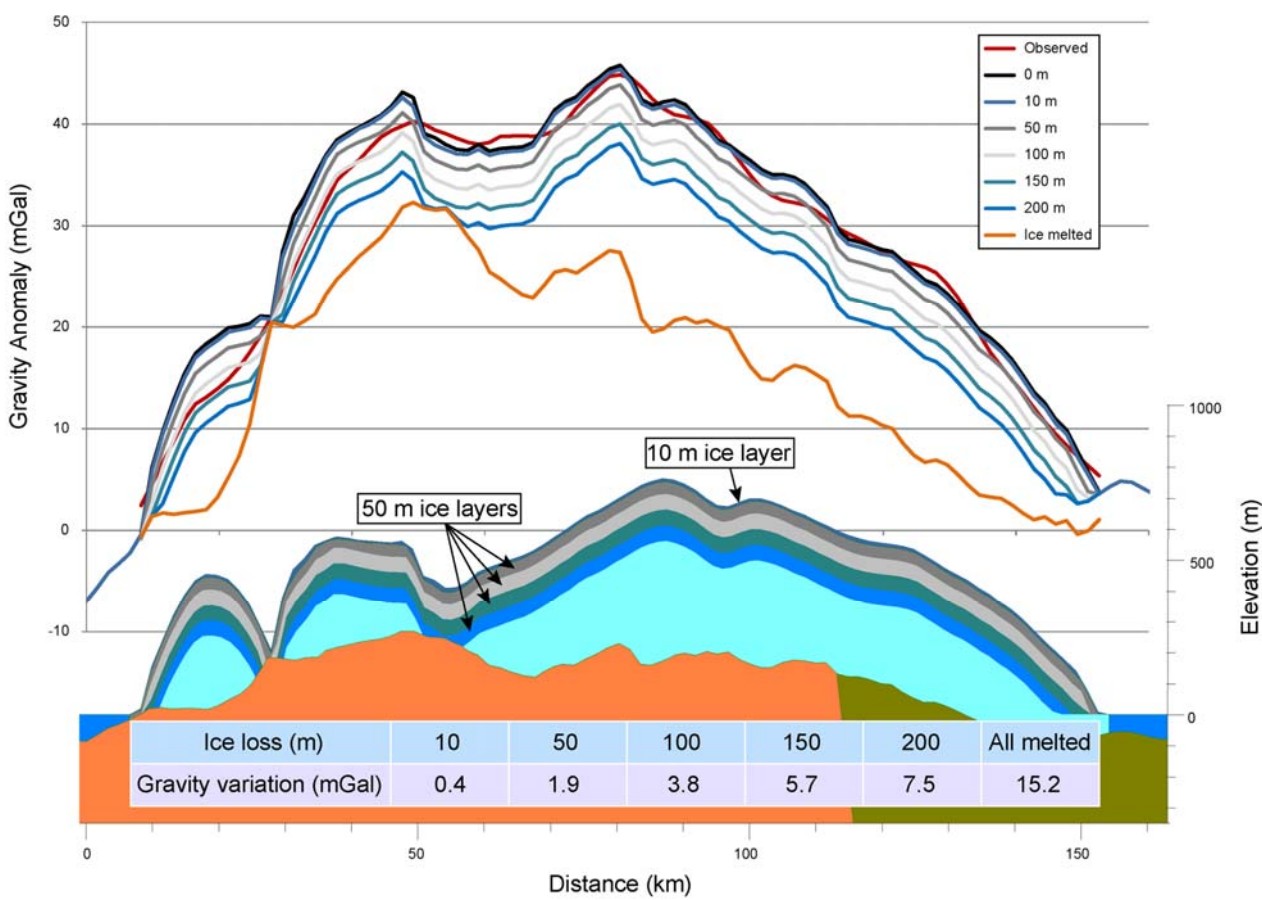

**Figure 8: Predicted gravity signature variations with ice loss. The gravity response is calculated using as the initial state the 2D-forward model from Profile A with the current known ice thickness. Uniform layers of ice of thickness of 10m, 50m, 100 m, 150 m and 200 m are removed from the model. The first 10 m-layer of ice loss yields to a gravity anomaly of approximatively 0.5 mGal. Significant ice loss is detectable from long-term observations.**

| Compilation | Magnetic | Gravity |
|---|---|---|
| Line Spacing | 4-8 km | 18 km |
| Aircraft altitude (approx.) | 900 m | 1000 m |
| Grid resolution | 2 km | 4 km |
| Acquisition | 1989-1991 | 1998-1999 |
| Acquired by | Sevmorgeo Amarok/TGS | Statens Kartverk Kort & Matrikelstyrelsen Universitet i Bergen |

**Table 1: Survey acquisition parameters of the magnetic and gravity compilation.**