# Peer review of "Revisiting Austfonna, Svalbard with potential field methods – A new characterization of the bed topography and its physical properties"

_The Cryosphere, 2019_

## Referee Comment (RC1) · Anonymous Referee #1 · 21 May 2019

General comments:

In summary, because of the importance of bed topography and sparse direct measurements of glaciers, the authors re-characterized the Austfonna, Svalbard bed topography and its physical properties with airborne gravity and magnetic measurements.

Overall, I have several questions for this manuscript, first, you revised the bed topography with gravity survey (section 4), however, there are lots of limitations and lack of analysis to figure out, whether the gravity method would be a good way to this study.

[Figure]

We don't know what the resolution of GPR/RES measurements is, and how good will be the gravity method. Based on your current presentation, I was not convinced. Second, since you picked two gravity profiles to do 2D forward models with them. I knew these two profiles chosen due to their location and coverage, however, do you think more 2D gravity (magnetic) forward models would help for the bed lithology reconstruction? Third, based on the profile A and B 2D forward models results (Figure 5), the bed topography from GPR/RES, corrected for observed gravity and the model elevation is quite different, which one is more reliable?

Specific comments: P1, line23-34: more and recent references should add in this paragraph. P2, line10-11: studies using gravity and magnetic data on glaciers are more than these. P3, line19: IGRF-> The International Geomagnetic Reference Field (IGRF). P4, line25: what is the resolution of this bed topography grid? P4, line33-35: In the later manuscript, P7, you said 100 m ice thickness variation is about 0.5 mGal gravity changes, so how the resolution and accuracy of the ice surface topography's influence the misfit? P5, line17-27: A statistical analysis of bed elevation between GPR/RES, corrected by observed gravity and 2D forward model results would be helpful to interpret the results. P6, line14: "susceptibility 0.006 SI in a 0.003 SI surrounding"->"0.004 SI"? typo? P6, line16: "(0.018 SI and 2730 km m-3)"-> "2750 (2725) km m-3 "? typo? P7, line24-25: how to understand this sentence? "The magnetic and gravity interpretation, having been flown in a grid pattern, are less sensitive to gridding interpolation . . .". P7, line 26: Can you explain, how did you get this 100 m ice thickness variation is ∼0.5 mGal variation in gravity? P7, line 34: reference? And also, the resolution of airborne gravity measurements also depends on the gravimeter, which was not discussed in this manuscript.

Figure1. Contour lines for elevation and thickness would be much more helpful than the color bars. Figure2. It would be better to represent all the gravity survey lines in this figure to make the choice of profile A & B much clearer. Figure5(b). There is no radar topography gravity response in the gravity panel for Profile B.

[Figure]

---

## Referee Comment (RC2) · Anonymous Referee #2 · 23 May 2019

This paper addresses an important and interesting topic regarding the estimation of sub-glacial bed topography from combined geophysical methods: combination of airborne gravity measurements with magnetic data. The authors compare their results to previous estimations obtained from GPR measurements. They found clear discrepancies in area with previous poor coverage and provide improvements in the overall estimation of the bedrock topography. An important improvement also concerns a fine description of the physical properties of the bedrock (lithology, roughness, density, thermal conductivity. . .) which are of prime interest to describe and predict the dynamics

of the ice cap.

This paper has been carefully prepared, is well written with figures of high quality. The context of the study is well presented. The readers will certainly appreciate the detailed analysis provided regarding the multiple geophysical methods used with systematic estimation of their uncertainties and discussion of their limitations. The geological interpretation presented by the authors is rigorous (I am not familiar with the area though), supported by both geophysical data and field observations. I only found that a more detailed discussion on the implications of having such improved estimations of sub-glacial bed topography on our understanding of the dynamics and evolution of the ice cap is missing. I would recommend to provide a dedicated paragraph with perspectives. In the following I raise few additional minor points.

Pg1, Line 29: The distribution of rheological properties of the ice might also play a critical role. Pg6, Line 13: typo Pg6, Line 31: Provide mean and std values of density for granites from the literature? Pg6, Line 40: Please provide more arguments to justify the link of this granite intrusion with the observed Caledonian Rijpfjorden. Pg 7, Line 41: What do you mean by "Calcite precipitation influences the bed roughness and the water film[...]". Please clarify. Pg8, Line 9: Do you have local evidences/estimations of this heat flux? Figure 4: Same color scale for all panels. Figure 5: Radar topography is missing in profile B. Some captions could be improved by providing more information with major observation/interpretation/messages (i.e. Figures 2, 3 and 6).

---

## Author Comment (AC1) · 20 Jun 2019

**Response to Review #1**

We thank anonymous referee #1 for his/her insightful and constructive comments. Below, the referee's comments (written in bold) are addressed with a response. We revised the manuscript accordingly.

**General comments:**

**In summary, because of the importance of bed topography and sparse direct measurements of glaciers, the authors re-characterized the Austfonna, Svalbard bed topography and its physical properties with airborne gravity and magnetic measurements. Overall, I have several questions for this manuscript, first, you revised the bed topography with gravity survey (section 4), however, there are lots of limitations and lack of analysis to figure out, whether the gravity method would be a good way to this study. We don't know what the resolution of GPR/RES measurements is, and how good will be the gravity method. Based on your current presentation, I was not convinced.**

We present a method that integrates gravity, magnetic and GPR/RES data which is more robust and reliable than using a single dataset. We did not acquire nor process the GPR/RES data and cannot assess the resolution of the measurements. However, we can comment on its spatial resolution which is non-uniform over the entire area due the geospatial distribution of the GPR/RES measurements. We modified in the bed topography revisited section:

"*Dunse et al. (2011) have presented a bedrock topography compilation from data acquired by RES and GPR.*"
to
"*Dunse et al. (2011) have presented a bedrock topography compilation with 1 km spatial grid resolution from data acquired by RES and GPR, but the geospatial distribution of the measurements (Fig.2) suggest lower resolution in areas with poor coverage.*"

The spatial resolution of the gravity data, here, is limited by the line spacing and the flight altitude. Improvement of the spatial resolution of the final bed topography would require denser line spacing for the gravity data. However, the focus of our study is to improve the accuracy and reliability of the bed topography by using a comparison along the gravity lines where data are most accurate and reliable. We added this remark in the methodology assessment section of the manuscript:

"*Therefore, using gravity modeling increases the confidence and the accuracy of the bedrock topography under a glaciated area.*"
to
"*Therefore, using gravity modeling increases the confidence and the accuracy of the bedrock topography under a glaciated area. Improvement of the spatial resolution of the final bed topography could also be achieved with the appropriate survey parameters and a denser line spacing for the gravity data.*"

**Second, since you picked two gravity profiles to do 2D forward models with them. I knew these two profiles chosen due to their location and coverage, however, do you think more 2D gravity (magnetic) forward models would help for the bed lithology reconstruction?**

We present the first combined gravity and magnetic model study on glaciated bed which help to reconstruct the bed lithology below the gravity lines. While the method has the potential to reconstruct the full bed lithology, the current dataset does not allow a full reconstruction given the

too wide line spacing and high flight altitude. We added this remark in the methodology assessment section:

"*These findings enhance the understanding of the regional geology of the area.*"
to
"*These findings enhance the understanding of the regional geology of the area and demonstrate the potential to reconstruct the full bed lithology with the aid of high-resolution gravity and magnetic data.*"

And we added in the conclusion section:

"*Higher resolution data from state-of-the-art instrumentation would further refine the physical properties of the basement.*"
to
"*Higher resolution data from state-of-the-art instrumentation would further refine the physical properties of the basement and allow a full reconstruction of the bed lithology and topography.*"

**Third, based on the profile A and B 2D forward models results (Figure 5), the bed topography from GPR/RES, corrected for observed gravity and the model elevation is quite different, which one is more reliable?**

The 2D-forward model is more reliable as it combines information from gravity, magnetic and GPR/RES.
Each separate dataset has limitations, but used together, they improve the overall knowledge of the glacier bed. In this present case, the GPR may have a high resolution where it has been densely acquired, but it shows reliability issues where the data are scarce. There are also technical difficulties when the GPR method is used with wet ice which can be expected on Svalbard. Adding gravity data offers a first level of correction based on the assumption that the bed lithology is homogeneous. The bed topography corrected for the 2D-forward model includes susceptibility and density parameters, taking account of the heterogeneity of the lithology. The latter model is therefore more reliable and adapted to the bed lithology of the studied area.

We clarified figure 5 by clearly demarking the bed topography corrected with 2D-forward and we added this caption: The gravity corrected bed topography provides a first level of correction but fails to recognize the heterogeneity of the bed. The 2D-forwad model improves the accuracy of the bed topography by using a density more representative of the lithology.

**Specific Comments:**
**P1, line23-34: more and recent references should add in this paragraph.**

We extended our literature references and included more examples.
L25: *Vaughan (IPCC) et al., 2013; Dowdeswell et al., 1997*
L26: *Pryzlibski et al., 2018; Grinsted, 2013; Radic et al., 2013; Barh et al., 2015*
L28: *Clarke, 2005*
L30: *e.g. Clarke, 2005*
L31: *Iverson et al. 2007; Eyles et al., 2015; Bamber et al. 2006*

**P2, line10-11: studies using gravity and magnetic data on glaciers are more than these.**

We agreed that there are more studies of gravity and magnetic data on glaciers, but we wanted to emphasize on few examples of magnetic and gravimetric interpretation, and 2D-forward model, of basement lithology studies in the polar regions. As far as we know, there are no 2D-forward model study under glaciers. However, we conducted a new literature search to include more example of gravity and magnetic studies. We modified:

*"Gravity and magnetic methods have been used independently in the past for basement studies in the Arctic and other glaciated areas (e.g. Gernigon et al., 2018; Døssing et al., 2016; Gourlet et al., 2015; Nasuti et al., 2015; Gernigon and Brönner, 2012; Olesen et al., 2010; Barrère et al., 2009; Spector, 1966)."*
to
*"Gravity and magnetic methods have been used in the past for basement lithology studies in the Arctic (e.g. Gernigon et al., 2018; Døssing et al., 2016; Nasuti et al., 2015; Gernigon and Brönner, 2012; Olesen et al., 2010; Barrère et al., 2009) and for sea-ice and glacier studies (An et al., 2017; Gourlet et al., 2015; Tinto et al., 2015; Zhao et al., 2015; Porter et al., 2014; Tinto et al., 2011; Studinger et al., 2008, Studinger et al., 2006; Spector, 1966)."*

**P3, line19: IGRF-> The International Geomagnetic Reference Field (IGRF).**

We modified the text accordingly.

**P4, line25: what is the resolution of this bed topography grid?**

The reliability and accuracy are increased along the 2D-lines. However, the spatial resolution of the gravity corrected bed topography remains the same as the gravity data. We clarified this in the text.

**P4, line33-35: In the later manuscript, P7, you said 100 m ice thickness variation is about 0.5 mGal gravity changes, so how the resolution and accuracy of the ice surface topography's influence the misfit?**

We agree that the resolution and accuracy of the ice surface topography influence the calculation of the bed elevation. However, in this study, both GPR depth measurements and gravity ice thickness were calculated with the same ice surface topography dataset which acts as a control variable. We added this remark to the text.

**P5, line17-27: A statistical analysis of bed elevation between GPR/RES, corrected by observed gravity and 2D forward model results would be helpful to interpret the results.**

The difference of the bed topography from the GPR/RES and the 2D-forward model varies between -170 m to 80 m with a standard deviation of 40 m. A smaller level of correction is required than predicted by the correction from the gravity solely since the 2D-forward model accounts for a certain degree of confidence on the GPR/RES data and for the bedrock density variation. We added this statistical analysis and this remark to the text.

**P6, line14: "susceptibility 0.006 SI in a 0.003 SI surrounding"->"0.004 SI"? typo?**

Yes, it should be written *"susceptibility 0.004 SI in a 0.003 SI surrounding"*. We modified the text accordingly.

**P6, line16: "(0.018 SI and 2730 km m-3)"-> "2750 (2725) km m-3 "? typo?**

Yes, it should be written *"(0.0018 SI and 2750 kg m$^{-3}$)"*. We modified the text accordingly.

**P7, line24-25: how to understand this sentence? "The magnetic and gravity interpretation, having been flown in a grid pattern, are less sensitive to gridding interpolation…".**

We regret that we have not been able to express ourselves in a clear manner. We meant that the coverage of the gravity and magnetic is consistent and regular all over the area while the GPR measurements distribution is irregular and therefore more prone to gridding interpolation artefacts. We reformulated this in the text.

**P7, line 26: Can you explain, how did you get this 100 m ice thickness variation is ~0.5 mGal variation in gravity?**

This sentence should read 10 m ice thickness variation is ~0.5 mGal. We conducted an ice loss model to establish for Austfonna the impact of an ice thickness variation. We removed uniform layers of 50m of ice by iteration to derive the theoretical gravity response from the ice loss. To clarify this comment, we now include a figure of the model in the manuscript with a brief description of the experiment. We modified:

*"Ice thickness variation of 100 m causes a variation in gravity of ~0.50 mGal which is resolved by state-of-the-art gravity measurements."*
to
*"Using Austfonna bed topography and lithology derived from the 2D-forward model, the theoretical gravity response was modeled for ice loss by removing iteratively uniform and homogeneous layers of ice (Fig.8). The model predicts that an ice thickness variation of 10 m causes an average variation in gravity of ~0.5 mGal which is resolved by state-of-the-art gravity measurements."*

**P7, line 34: reference? And also, the resolution of airborne gravity measurements also depends on the gravimeter, which was not discussed in this manuscript.**

We agree that the spatial resolution of the airborne gravity measurements depends on the gravimeter as well as the platform, the line spacing, the acquisition speed and the altitude above the source. We modified the text accordingly with this remark and referenced to similar gravity studies (e.g. An et al., 2017; Studinger et al., 2008).

**Figure1. Contour lines for elevation and thickness would be much more helpful than the color bars.**

We added contour lines for the ice thickness as it is most relevant to the study. Adding the contour lines for elevation would have been confusing and too much information to be readable.

**Figure2. It would be better to represent all the gravity survey lines in this figure to make the choice of profile A & B much clearer.**

We have tried to do so, but the figure had too much information to be readable.

**Figure5(b). There is no radar topography gravity response in the gravity panel for Profile B.**

We did not recreate the experiment with Profile B as Profile A was sufficient to demonstrate the advantage of using gravity to correct the GPR measurements and largely discussed in the previous section. The radar gravity response is used in Profile A to explain our choice to modify the bed topography as no other bed density could explain the gravity signature.

---

## Author Comment (AC2) · 20 Jun 2019

**Response to Review #2**

We thank anonymous referee #2 for his/her positive and constructive comments. Below, the referee's comments (written in bold) are addressed with a response. We revised the manuscript accordingly.

**This paper addresses an important and interesting topic regarding the estimation of sub-glacial bed topography from combined geophysical methods: combination of airborne gravity measurements with magnetic data. The authors compare their results to previous estimations obtained from GPR measurements. They found clear discrepancies in area with previous poor coverage and provide improvements in the overall estimation of the bedrock topography. An important improvement also concerns a fine description of the physical properties of the bedrock (lithology, roughness, density, thermal conductivity…) which are of prime interest to describe and predict the dynamics of the ice cap.**
**This paper has been carefully prepared, is well written with figures of high quality. The context of the study is well presented. The readers will certainly appreciate the detailed analysis provided regarding the multiple geophysical methods used with systematic estimation of their uncertainties and discussion of their limitations. The geological interpretation presented by the authors is rigorous (I am not familiar with the area though), supported by both geophysical data and field observations.**

We are grateful that the referee appreciated our manuscript.

**I only found that a more detailed discussion on the implications of having such improved estimations of sub-glacial bed topography on our understanding of the dynamics and evolution of the ice cap is missing. I would recommend to provide a dedicated paragraph with perspectives.**

The referee raised an interesting point. The current gravity data does allow to improve the accuracy, but the resolution of the entire area is not yet high enough to make a robust 3D model of the bed topography. A study of the dynamics and evolution of the ice is currently beyond the focus of this manuscript.

**In the following I raise few additional minor points.**
**Pg1, Line 29: The distribution of rheological properties of the ice might also play a critical role.**

We agree with the referee. This has been added to the text.

**Pg6, Line 13: typo**

The text *"to the minimum size of the required for this depth"* has been modified to *"to the minimum size required for this depth"*.

**Pg6, Line 31: Provide mean and std values of density for granites from the literature?**

Granite densities are expected to vary between 2500 – 2810 kg m$^{-1}$ with an average of 2640 kg m$^{-1}$ (Tellford, 1990). The text now includes this information.

**Pg6, Line 40: Please provide more arguments to justify the link of this granite intrusion with the observed Caledonian Rijpfjorden.**

On the magnetic map with the geological background, the high anomalies correspond to the geological mapping of the observed granites as well as the susceptibility and density modeled suggest affinities with granites. We modified the text accordingly.

**Pg 7, Line 41: What do you mean by "Calcite precipitation influences the bed roughness and the water film[…]". Please clarify.**

As the glacial basal temperature is near the pressure-melting point, the water film, created through that process, causes the dissolution of calcite from the carbonate bed on the upglacier side of the bed obstacles. During the regelation on the lee side of the bed obstacles, the calcite is chemically precipitated. Subglacial dissolution and precipitation processes of the calcite on the bed regulate the calcite saturation of the water film and modify the bed morphology and roughness. We suggest modifying the text as:

"*Calcite dissolution and precipitation chemical processes have a role on the calcite saturation of the water film that lubricates the bed-glacier interface and modify the bed morphology and roughness through melting and regelation processes.*"

**Pg8, Line 9: Do you have local evidences/estimations of this heat flux?**

On Austfonna, only one borehole has been drilled to reach the bedrock and provide heat flux information in the summit area (Ignatieve and Macheret, 1991; Zagorodnov et al., 1989). There are no other direct observations available to estimate this heat flux. Our study suggests that this value may not reflect the entire bed underlying Austfonna. We modified the text with this comment.

**Figure 4: Same color scale for all panels.**

This has been modified accordingly.

**Figure 5: Radar topography is missing in profile B.**

We did not recreate the experiment with Profile B as Profile A was sufficient to demonstrate the advantage of using gravity to correct the GPR measurements and largely discussed in the previous section. The radar gravity response is used in Profile A to explain our choice to modify the bed topography as no other bed density could explain the gravity signature.

**Some captions could be improved by providing more information with major observation/interpretation/messages (i.e. Figures 2, 3 and 6).**

We modified the captions accordingly:
Figure 2: *The interpreted profiles have been chosen to cover a large area of Austfonna and to capture important geological trend.*

Figure 3: *The gravity data are sensitive to an excess or loss of mass. Lower gravity data are often linked to sedimentary basins. The magnetic data show important N-S trending anomalies crossing Nordaustlandet and intersecting with the Caledonian Rijpforden granites.*

[Figure]

Figure 6: *Sills bodies located on the North-East onshore and offshore are generally shallower than the large and deep NS trending granitic intrusions crossing Austfonna.*

---

## Author Response (AR1)

**Authors' response**

Dear Editor Daniel Farinotti,

We thank you for the comments and suggestions. Below, the editor's comments (written in bold) are addressed with a response. We revised the manuscript accordingly.

**Comments to the Author:**

**Dear authors,**
**Many thanks for posting your answers to the two referees that reviewed the manuscript.**

**As far as I can see, the only critical point raised by the reviewers (and particularly by Reviewer #1) is the discussion of the accuracy of the final bedrock topography, i.e. an assessment of the actual added value provided by the gravity measurements. Paraphrasing, the reviewer's first and third comment ask for a quantification of the accuracy of the final result, or better, a metric quantifying the impact (in terms of accuracy) that the use of gravity measurements has.**

**I think that this point is important, also because your main text stresses the fact that such an improvement is actually happening (e.g. second-to-last sentence in both your abstract and the introduction). At the moment, one might argue that this improvement is postulated, rather than shown.**

**I understand the difficulty in "showing" such an improvement, especially since reliable ground-truth information is difficult to obtain. However, I have the impression that the original GPR/RES data (i.e. the one by Downdeswell et al., 1986 [https://doi.org/10.3189/S0260305500001130], rather than the interpolated ones by Dunse et al., 2011) might be of help. Whilst I'm not sure how feasible it will be to reconstruct distributed reliability information for that data, I imagine that at least the information about the spatial distribution of the data (the interpreted dataset is available through the Glacier Thickness Database: https://www.gtn-g.ch/data_catalogue_glathida/) can be very useful in strengthening your discussion on the matter.**

**The remaining points raised by the reviewer's seem more straight-forward to handle, as your answers show. Reviewer #2 also comments on the importance of the geological interpretation, noticing, however, that a geologist more familiar with the region should provide his/her advice on it.**

**In light of the above, I don't see major obstacles for a final publication. However, I think it is appropriate having the amended version of your article seen again by an external referee (TC's system calls this a "major revision" which might sound unduly "harsh" in your case).**

**In this sense, I would like to ask you for such a new manuscript version to be produced and posted, so that the review of the amended version can happen.**

**Many thanks for your collaboration and best wishes.**
**Daniel Farinotti**

Response:

We followed your advices and looked at the raw data and their spatial distribution by using both the Glacier Thickness Database and the original contribution by Dowdeswell et al. (1986). One major technological improvement between the RES measurements in 1983, pre-GPS era, and the gravity data (2000) is the improvement of the accuracy and quality of the navigational instruments and positioning processing. The coverage from magnetic and gravity data is more uniform while the coverage of the RES is hindered by navigational error and poor GPS accuracy. We added a discussion about these errors related to the acquisition in the manuscript.

We also extended the discussion about the commonly known sources of errors in GPR processing. In particular, we addressed the presence of wet ice and water in a glacier. Austfonna is a polythermal glacier and is susceptible to error thickness due to the presence of water, whether it is from scattering or uncertainties on the radio-wave velocity assumption / time-to-thickness conversion. Using magnetic and gravity data combined with the GPR/RES data enhances the bed topography where the RES has a poor coverage and poor GPS accuracy and provides extra constraints in the possible the presence of water.

The new manuscript has also been modified accordingly to the comments and suggestions provided by the two referees (tc-2019-74-RC1.pdf and tc-2019-74-RC2.pdf). Additional grammatical mistakes and typos have been corrected. A copy of the manuscript with track changes is also available.

Best Regards,
Marie-Andrée Dumais & Marco Brönner

[revised manuscript text omitted]

---

## Author Response (AR2)

**Authors' response**

Dear Editor Daniel Farinotti,

We thank you for the comments and suggestions. We implemented the text accordingly. We also considered the comments from reviewer #3.

We could not provide an estimate of the uncertainty caused by water accumulation. Several parameters, such as the scattering and time-to-thickness conversions, would interfere with the measurements. We precise the text by mentioning that the presence of water would cause underestimation of the ice thickness.

We migrated our datasets in the public geophysical database of the Geological Survey of Norway (http://geo.ngu.no/GeosciencePortal/search). NGU policy is that our datasets and services are open, accessible and free for downloading. This approach secure usage of our data for civil purposes to the maximum extent. We develop standardized data deliveries and services for national and international management through active participation in Norway Digital and work with The Public Map Data (DOK). This work is in accordance with applicable standards and guidelines given in the Geodata Act and related regulations. The Geodata Act implements Directive 2007/2 / EC of 14 March 2007, establishing an Infrastructure for Spatial Information in the European Community (INSPIRE) as a directive in Norwegian law.

Many thanks for the help you have provided to us during the revision process.

Best regards,

Marie-Andrée Dumais and Marco Brönner